# Improving Sampling Distribution of Off-policy Training in Generative Flow Networks

## Abstract

Generative Flow Networks (GFlowNets) are a new family of amortized sampling methods that learn a distribution over compositional objects proportional to their rewards. GFlowNets showcases remarkable capabilities in generating diverse samples compared to reward-maximizing reinforcement learning (RL), mainly due to its off-policy training scheme with replay buffers. Specifically, several methods sample a batch of samples from the replay buffer proportional to their reward or loss values to promote sample efficiency or exploration. However, we discovered that prior methods face a critical but overlooked challenge, where they suffer from mode collapse or over-exploration on hard-to-learn samples with the current policy. To mitigate this issue, we define *learnability*, which can be measured by introducing a reference policy. We find that using learnability information to sample batches from the buffer during off-policy training significantly improves the efficacy of GFlowNets, especially when the reward landscape is sparse and has distinct modes. We validate the effectiveness of our approach on two variants of synthetic grids with exploration challenges, biochemical discovery, and combinatorial optimization tasks.

## 1. Introduction

Generative Flow Networks, or GFlowNets (Bengio et al., 2021; 2023), are a new family of amortized sampling methods that learn a distribution over compositional objects such as graphs or strings proportional to their rewards. In GFlowNets, we aim to learn a stochastic policy $\pi(x)$ that constructs objects from a sequence of actions. In terms of transforming sampling as a sequential decision-making

problem, GFlowNets resembles reinforcement learning (RL, Sutton et al., 1998), although standard RL focuses on reward-maximization while GFlowNets focuses on reward-matching property. GFlowNets' capability of discovering high-rewarding and diverse samples results in success on various domains, including molecule discovery (Bengio et al., 2021; Pandey et al., 2025), biological sequence design (Jain et al., 2022; Kim et al., 2025a), combinatorial optimization (Zhang et al., 2023; Kim et al., 2025b), and large language models (Hu et al., 2024; Zhu et al., 2025).

One of the advantages of GFlowNets is an off-policy training scheme. While most RL-based methods rely on on-policy methods (Schulman et al., 2017; Shao et al., 2024), GFlowNets utilize a replay buffer during training to improve mode coverage (Malkin et al., 2023). As we do not need to use the current policy as a behavior policy, several methods have been discussed for designing an effective exploratory policy during the on-policy sampling stage (Kim et al., 2025c; Madan et al., 2025; Malek et al., 2025).

A relatively unexplored region is how to design a sampling distribution from the replay buffer. The most widely used approach is uniform sampling distribution, i.e., samples trajectories in the buffer uniformly at random. Unfortunately, it may focus on low-rewarding regions and be hard to cover multiple modes when the search space is too large and the reward landscape is sparse. While some methods focus on high-rewarding samples (Shen et al., 2023; Vemgal et al., 2023), these approaches may lead to a severe mode collapse when certain modes have excessively high rewards.

One of the promising approaches is sampling trajectories proportional to their loss, similar to the PER (Schaul et al., 2015), suggested in the RL literature. While it leads a policy to explore unseen regions, we discover that relying solely on loss values incurs high variance in GFlowNets, as we only observe a single trajectory that falls into a certain terminal state $x$. Furthermore, if a certain state is too far away from most of the dataset we collected so far, it is hard to learn a correct policy from that sample, especially when the search space is large, as demonstrated in Figure 1.

To overcome the aforementioned challenges, we introduce a new term called *learnability*, which is a measure of how the observed sample is learnable with the current policy. To

[1]Anonymous Institution, Anonymous City, Anonymous Region, Anonymous Country. Correspondence to: Anonymous Author <anon.email@domain.com>.

Preliminary work. Under review by the International Conference on Machine Learning (ICML). Do not distribute.

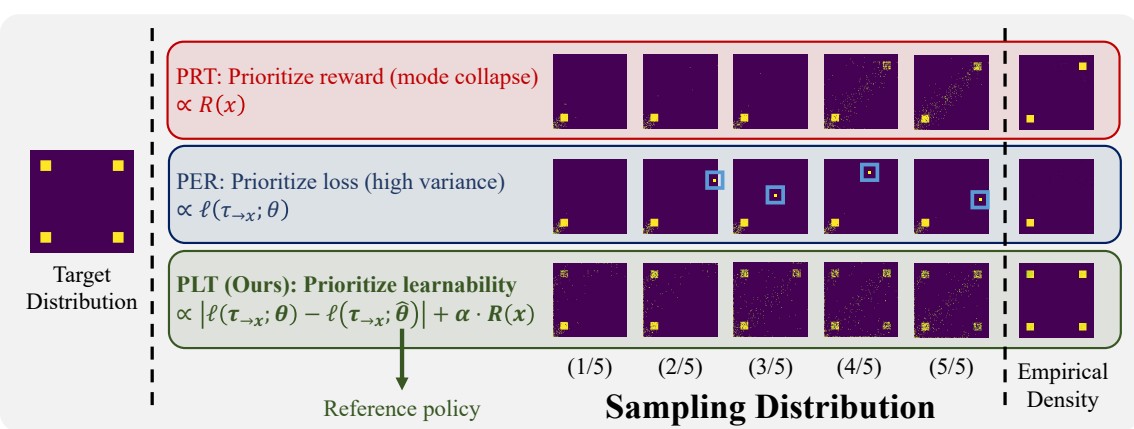

*Figure 1.* Motivating Figure. In off-policy training of GFlowNets, naively prioritizing reward or loss values might lead to mode collapse and unstable training, particularly in sparse rewards and long horizons. By introducing learnability, we progressively learn to cover multiple modes in the target distribution.

measure learnability, we prepare a reference policy, which is a lagged version of the current policy, and compute the difference between the loss from the current policy and the reference policy. There are mainly two circumstances where we prioritize the sample: (1) low loss for the current policy and high loss for the reference policy, and (2) high loss for the current policy and low loss for the reference policy. For the former, we can prioritize recently acquired samples from the buffer, and for the latter, we can prioritize samples that are forgotten by the current policy. By prioritizing those samples, we can prevent the policy from mode collapse and wasted updates of its parameters on unlearnable outliers.

We conduct experiments on two variants of a synthetic grid with exploration challenges, as well as four biochemical discovery tasks and two combinatorial optimization tasks. We validate that our method achieves better performance compared to several relevant baselines. We also show that our method is robust to different GFlowNets objectives and exploratory policies suggested in literature.

## 2. Preliminaries

### 2.1. Generative Flow Networks

Generative Flow Networks, or GFlowNets (Bengio et al., 2021; 2023), are a class of probabilistic generative models designed to sample compositional objects $x \in \mathcal{X}$ with probability proportional to a non-negative reward function, $R(x)$. Instead of sampling $x$ with a single forward pass, GFlowNets follow a constructive generative process: objects are built sequentially through discrete *actions* that modify an intermediate *state*. The entire construction process can be represented as a directed acyclic graph (DAG) $G = (\mathcal{S}, \mathcal{A})$, where $\mathcal{S}$ denotes the state space (nodes) and $\mathcal{A}$ denotes the action space (edges). A unique initial state $s_0 \in \mathcal{S}$ has no incoming edges, and a set of terminal states $\mathcal{X} \subset \mathcal{S}$ has no outgoing edges. A complete sequence

$\tau_{\to x} = (s_0 \to s_1 \to \cdots \to s_T = x)$, called a *trajectory*, corresponds to one way of constructing $x$.

To model this process, GFlowNets learn stochastic *policies* that define transition probabilities over edges in the DAG. The forward policy $P_F(s'|s; \theta)$ specifies how to move from a state $s$ to one of its children $s'$, thereby inducing the trajectory probability

$$P_F(\tau; \theta) = \prod_{t=1}^{T} P_F(s_t \mid s_{t-1}; \theta). \quad (1)$$

The backward policy $P_B(s|s')$ analogously defines probabilities over parents of a state and allows backward traversal of trajectories. The training objective of GFlowNets is to align the marginal probability of generating a terminal state with its reward, i.e.,

$$P_F^T(x; \theta) = \sum_{\tau_{\to x}} P_F(\tau; \theta) \propto R(x). \quad (2)$$

**Objectives.** In practice, $P_F$ (and sometimes $P_B$) are parameterized by neural networks and trained with various objectives to satisfy Eq. (2). We briefly introduce two widely used objectives, trajectory balance (TB, Malkin et al., 2022) and detailed balance (DB, Bengio et al., 2023).

**TB** introduces a total flow $Z_\theta$ to approximate the partition function. Given a trajectory $\tau = (s_0 \to \cdots \to s_T = x)$, TB aims to minimize the loss in Eq. (3).

$$\ell(\tau_{\to x}; \theta) = \left( \log \frac{Z_\theta \prod_{t=1}^{T} P_F(s_t|s_{t-1}; \theta)}{R(x) \prod_{t=1}^{T} P_B(s_{t-1}|s_t; \theta)} \right)^2. \quad (3)$$

**DB** considers flow matching at the edge level instead of the trajectory level and introduces state flow function $F_\theta : \mathcal{S} \to \mathbb{R}_{\geq 0}$, which approximates the total flow through state $s$. Given an intermediate transition $(s_{t-1} \to s_t)$, DB aims

to minimize the loss in Eq. (4), with $F_\theta(s_t)$ replaced by the terminal reward $R(x)$ at terminal states for $t = T$ (i.e., $F_\theta(x) = R(x)$).

$$\ell(s_{t-1}, s_t; \theta) = \left( \log \frac{F_\theta(s_{t-1}) P_F(s_t | s_{t-1}; \theta)}{F_\theta(s_t) P_B(s_{t-1} | s_t; \theta)} \right)^2 . \quad (4)$$

### 2.2. Prior approaches for off-policy training

Several approaches have been proposed for improving the off-policy training of RL and GFlowNets. We briefly introduce two widely used techniques in the replay buffer, prioritized replay training (PRT) and prioritized experience replay (PER).

In **PRT**, we prioritize high-rewarding samples to improve the sample efficiency of the policy training. In other words, we sample terminal state $x$ proportional to its reward, $R(x)$, as follows:

$$x \sim P_\mathcal{B}(x) \propto R(x). \quad (5)$$

To balance exploration and exploitation, Shen et al. (2023) proposed a more advanced approach, which samples $50\%$ from the top 10 percentile of $R(x)$, and samples the rest from the bottom 90 percentile. From now on, we will refer to this approach as a PRT.

In **PER**, we prioritize high-loss samples to improve the exploration of the off-policy training. As we only have access to loss of trajectory $\ell(\tau_{\to x}; \theta)$ instead of marginal one $\ell(x; \theta) = \mathbb{E}_{\tau_{\to x}}[\ell(\tau_{\to x}; \theta)]$, we sample terminal state $x$ proportional to the trajectory loss as follows:

$$x \sim P_\mathcal{B}(x) \propto \ell(\tau_{\to x}; \theta). \quad (6)$$

Note that once we sample the terminal state $x$ from the buffer for training, we reconstruct trajectories using a backward policy $P_B$, compute the loss in trajectory level, and update the loss information in the buffer.

## 3. Methodology

In this section, we introduce the **PLT** (**P**rioritized **L**earnability **T**raining), which is a novel approach for improving the sampling distribution of off-policy training in GFlowNets based on learnability.

### 3.1. Measuring learnability

GFlowNets aim to train a stochastic policy that matches the reward function, rather than maximizing the reward. To enable reward matching, it is crucial to utilize a replay buffer during training. The most widely used approach for promoting exploration is the PER approach, which samples terminal states in proportion to their loss values. However, as the trajectory length becomes longer, the variance of the

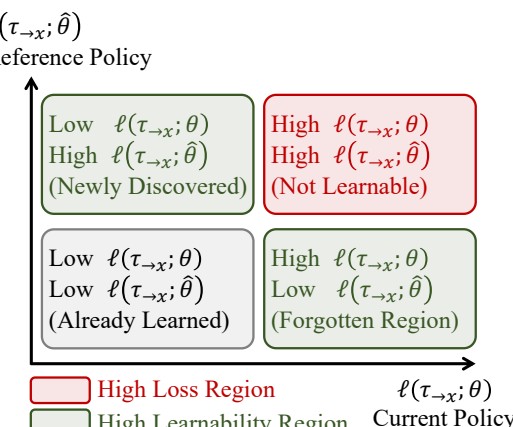

*Figure 2.* PER naturally prioritizes samples with very high loss (probably high loss for the reference policy). Focusing on learnability prioritizes samples that are newly discovered or forgotten by the current policy.

observed loss naturally increases. This happens since we can only observe a single trajectory that falls into a terminal state $x$ to compute the loss. For off-policy training, we sample a terminal state $x$ from the buffer and reconstruct a trajectory with the backward policy $P_B$. When the state space is huge, most intermediate states in reconstructed trajectories are totally unseen by the current policy $P_F$, which makes learning unstable. These limitations motivate the need for a more informed sampling distribution that can guide efficient off-policy training of GFlowNets.

To select samples from the replay buffer wisely, we introduce a new measure called *learnability*, which indicates how much information can be provided by the selected terminal state to the current policy. To measure learnability, we first define a *reference policy*, $P_F(\cdot | \cdot; \hat{\theta})$, a lagged version of the current policy. Based on the reference policy, we define learnability as follows:

$$\text{Learnability}(x) = |\ell(\tau_{\to x; \theta}) - \ell(\tau_{\to x; \hat{\theta}})|. \quad (7)$$

As depicted in Figure 2, exploiting learnability means that we concentrate on both recently discovered regions (high $\ell(\tau_{\to x; \hat{\theta}})$ and low $\ell(\tau_{\to x; \theta})$) and forgotten regions (low $\ell(\tau_{\to x; \hat{\theta}})$ and high $\ell(\tau_{\to x; \theta})$), while preventing to concentrate on samples that are too far away to learn with the current policy.

### 3.2. Improving sampling distribution of off-policy training

Based on learnability, we design an improved sampling distribution of off-policy training, PLT. The role of PLT is to select samples from the replay buffer to efficiently improve the current policy, thereby covering a multi-modal distribution. To this end, our proposed approach is to sample a terminal state proportional to the mixture of learnability

and the reward as follows:

$$x \sim P_{\mathcal{B}}(x) \propto |\ell(\tau_{\to x}; \theta) - \ell(\tau_{\to x}; \hat{\theta})| + \alpha \cdot R(x). \quad (8)$$

This approach encourages the GFlowNet policy to sample regions with both high learnability and high reward. Here, $\alpha$ is the mixing coefficient that trades off between learnability and the reward. We investigate the robustness of our method on $\alpha$ in Section 5.3.

**Implementation of PLT.** Our method can play a plug-and-play module for various existing GFlowNet training algorithms in an orthogonal way. As presented in Algorithm 1, PLT only introduces periodical reference policy initializations and $\ell(\tau_{\to x}; \hat{\theta})$ computations compared to the PER approach, which does not cost a significant time. We investigate the effect of the reference policy update interval $M$ in Section 5.3 and the additional time complexity of our method in Section A.4.

---

**Algorithm 1** PLT

1: **Input:** Max rounds $T$; Sampling batch size $b_1$; Training batch size $b_2$; Update interval $M$; Mixing coefficient $\alpha$; Reward function $R$.
2: Initialize policy $P_F(\tau; \theta)$ and buffer $\mathcal{B} \leftarrow \emptyset$
3: Initialize reference policy $\hat{\theta} \leftarrow \texttt{deepcopy}(\theta)$
4: **for** $t = 0, \ldots, T-1$ **do**
5:    // On-policy Sampling
6:    Sample trajectories $\tau_1, \cdots, \tau_{b_1}$ from $P_F(\tau; \theta)$
7:    Compute rewards: $R(x_1), \cdots, R(x_{b_1})$
8:    Compute loss with $\theta$: $\ell(\tau_1; \theta), \cdots, \ell(\tau_{b_1}; \theta)$
9:    Compute loss with $\hat{\theta}$: $\ell(\tau_1; \hat{\theta}), \cdots, \ell(\tau_{b_1}; \hat{\theta})$
10:   Update $\mathcal{B} \leftarrow \mathcal{B} \cup \{(x_i, R(x_i), \ell(\tau_i; \theta), \ell(\tau_i; \hat{\theta})\}_{i=1}^{b_1}$
11:
12:   // Off-policy Training
13:   Sample $x_1, \cdots, x_{b_2}$ from $\mathcal{B}$ proportional to:
14:   $|\ell(\tau_{\to x}; \theta) - \ell(\tau_{\to x}; \hat{\theta})| + \alpha \cdot R(x)$
15:   Generate trajectories $\tau_1, \cdots, \tau_{b_2}$ with $P_B$
16:   Update $\theta$ with to minimize $\frac{1}{b_2} \sum_{i=1}^{b_2} \ell(\tau_i; \theta)$
17:   **if** $t \bmod M = 0$ **then**
18:     Update reference policy $\hat{\theta} \leftarrow \texttt{deepcopy}(\theta)$
19:   **end if**
20: **end for**

---

# 4. Experiments

In this section, we present extensive experimental results to validate the superiority of our method on various benchmarks. We conduct experiments on two gridworld environments, four biochemical tasks, and two combinatorial optimization tasks. Our code is available at this link.

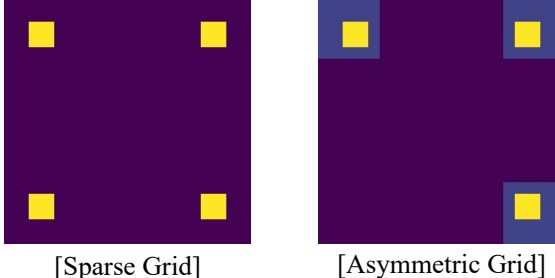

[Sparse Grid]      [Asymmetric Grid]

*Figure 3.* Visualization of target distributions of sparse and asymmetric grid in $d = 2$, $H = 128$.

## 4.1. Gridworld

We first evaluate PLT in a synthetic grid environment introduced by Bengio et al. (2021) with more challenging scenarios, a sparse and asymmetric grid. The environment basically consists of a $d$-dimensional hypercube of side length $H$, which results in a search space of size $\mathcal{O}(H^d)$. The agent always starts from the origin state ($s_0 = (0, \cdots, 0)$), and actions are operations that increase one of the coordinates in a state by 1 while not exiting the grid, or terminate to receive the reward. For the sparse grid, the reward function is defined as:

$$R(x) = R_0 + R_2 \prod_{i=1}^{d} \mathbb{I}\left[|\tilde{x}^i| \in (0.3, 0.4)\right], \quad (9)$$

where $\tilde{x}^i = \frac{x^i}{H-1} - 0.5$. For the asymmetric grid, the reward function is defined as:

$$
R(x) = R_0 + R_1 \left[\prod_{i=1}^{d} \mathbb{I}\left[|\tilde{x}^i| > 0.25\right] - \prod_{i=1}^{d} \mathbb{I}\left[\tilde{x}^i < -0.25\right]\right] \\
+ R_2 \left[\prod_{i=1}^{d} \mathbb{I}\left[|\tilde{x}^i| \in (0.3, 0.4)\right] - \prod_{i=1}^{d} \mathbb{I}\left[\tilde{x}^i \in (-0.4, -0.3)\right]\right],
$$
$$(10)$$

where $R_0 = 1e-5$, $R_1 = 0.1$, and $R_2 = 2.0$. In a sparse grid, there are no regions with $R_1$, so it requires more exploration power to cover all modes. In an asymmetric grid, we remove the mode nearest to the initial state, requiring exploration at the early stage. We visualize the target reward distributions in Figure 3.

**Baselines.** We compare our method with on-policy TB (**On-Policy**) and several off-policy techniques, including uniform sampling (**Uniform**), reward-prioritized (**PRT**), and loss-prioritized (**PER**) sampling from the buffer. Note that all hyperparameters and network configurations are fixed across methods to isolate the effect of the sampling distribution of off-policy training. Please refer to Section A.1 for more details of the experimental setup.

*Table 1.* Main experiment results in gridworld. We report the L1 density error of different approaches of sampling transitions from the buffer during off-policy training. Experiments are conducted with five different seeds.

| Method | $d=2, H=128$ $(\times 10^{-5})$ | | $d=2, H=256$ $(\times 10^{-5})$ | | $d=3, H=64$ $(\times 10^{-6})$ | | $d=4, H=48$ $(\times 10^{-7})$ | |
|---|---|---|---|---|---|---|---|---|
| | Sparse (S) | Asymmetric (A) | Sparse (S) | Asymmetric (A) | Sparse (S) | Asymmetric (A) | Sparse (S) | Asymmetric (A) |
| On-Policy | $9.156 \pm 0.000$ | $4.307 \pm 0.091$ | $2.319 \pm 0.000$ | $1.783 \pm 0.190$ | $6.681 \pm 0.000$ | $6.362 \pm 0.409$ | $3.678 \pm 0.124$ | $3.538 \pm 0.002$ |
| Uniform | $2.434 \pm 3.752$ | $2.982 \pm 1.604$ | $0.820 \pm 0.060$ | $0.543 \pm 0.067$ | $1.089 \pm 0.084$ | $2.285 \pm 0.743$ | $3.165 \pm 0.714$ | $2.211 \pm 0.024$ |
| PRT | $3.988 \pm 3.608$ | $5.377 \pm 3.787$ | $1.163 \pm 0.706$ | $0.521 \pm 0.068$ | $0.890 \pm 0.127$ | $4.751 \pm 1.609$ | $0.578 \pm 0.021$ | $2.454 \pm 0.237$ |
| PER | $4.186 \pm 4.538$ | $1.634 \pm 1.357$ | $1.378 \pm 0.264$ | $0.599 \pm 0.075$ | $0.845 \pm 0.063$ | $2.108 \pm 0.794$ | $\mathbf{0.576 \pm 0.025}$ | $2.237 \pm 0.029$ |
| PLT (Ours) | $\mathbf{0.729 \pm 0.053}$ | $\mathbf{1.053 \pm 0.092}$ | $\mathbf{0.542 \pm 0.162}$ | $\mathbf{0.510 \pm 0.033}$ | $\mathbf{0.719 \pm 0.068}$ | $\mathbf{1.824 \pm 0.133}$ | $0.597 \pm 0.038$ | $\mathbf{2.195 \pm 0.024}$ |

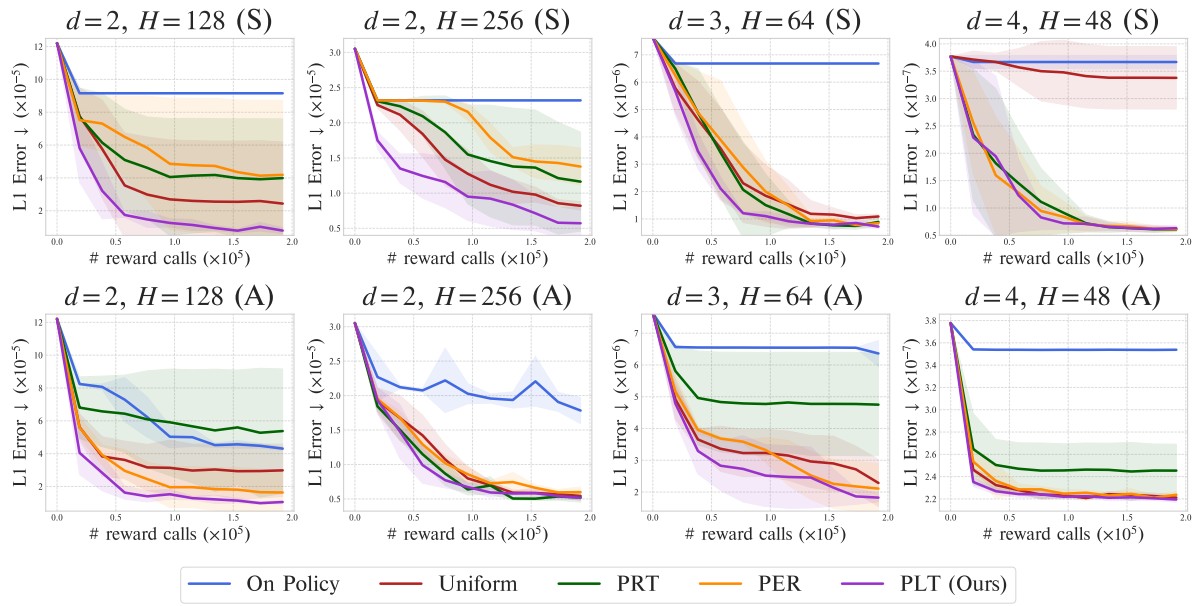

*Figure 4.* L1 density error over the course of training in gridworld environments. Experiments are conducted with five different seeds. We report the number of discovered modes in Section B.1.

**Evaluation Results.** We compare our method against baselines in terms of empirical L1 distance between the target distribution and the empirical distribution from the trained policy. As shown in Table 1, our method mostly outperforms the baselines in gridworld experiments with diverse dimensions and side lengths. Notably, the gap between our method and other baselines becomes substantially larger as the horizon lengthens, which aligns with our initial motivation: the high variance of loss values. To better understand learning dynamics, Figure 4 shows the evolution of evaluation metrics over training. As depicted in the figure, our method not only achieves the lowest L1 density error but also demonstrates rapid convergence, indicating the effectiveness of our sampling distribution design. In contrast, PRT exhibits high variance, indicating that it is highly vulnerable to the mode collapse. We also report the estimation error of the log partition function ($\log Z$) in **??** to analyze learning stability.

### 4.2. Biological and Chemical Discovery

To validate the effectiveness of our method in real-world tasks, we conduct experiments on biochemical tasks. In these tasks, we sequentially construct molecules or biological sequences by adding atoms or fragments. Our objective is to discover *modes*, high-rewarding samples that are distinct from other samples based on a pre-defined similarity constraint. In biochemical tasks, discovering multiple distinct modes is crucial for the robustness to proxy misspecification (Bengio et al., 2021). We conduct experiments on four benchmarks: QM9, TFBind8, sEH, and L14-RNA, following Shen et al. (2023). Please refer to Section A.2 for more details of the experimental setup.

**QM9:** Generate a small molecular graph consisting of 5 blocks that maximizes the HOMO-LUMO gap on the target transcription factor, which is obtained via a pre-trained MXMNet proxy from Zhang et al. (2020).

**sEH:** Generate a small molecular graph consisting of 6 blocks that maximizes the binding affinity to soluble epoxide hydrolase (sEH), which is provided by the pre-trained proxy model from Bengio et al. (2021).

**TFBind8:** Generate a DNA sequence with 8 nucleotides that maximizes a binding affinity to a human transcription factor, which is provided by a pre-trained proxy model from Trabucco et al. (2022).

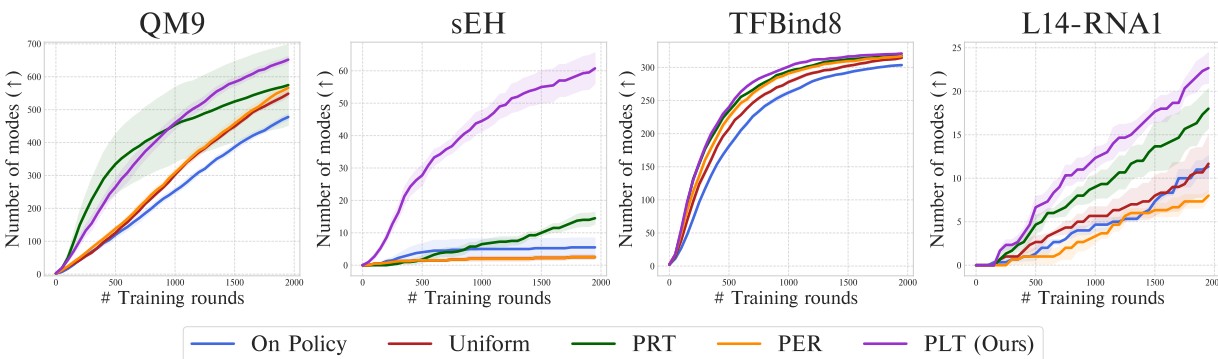

*Figure 5.* Main experiment results in biochemical tasks. We report the number of modes discovered over the course of training. Experiments are conducted with four different seeds.

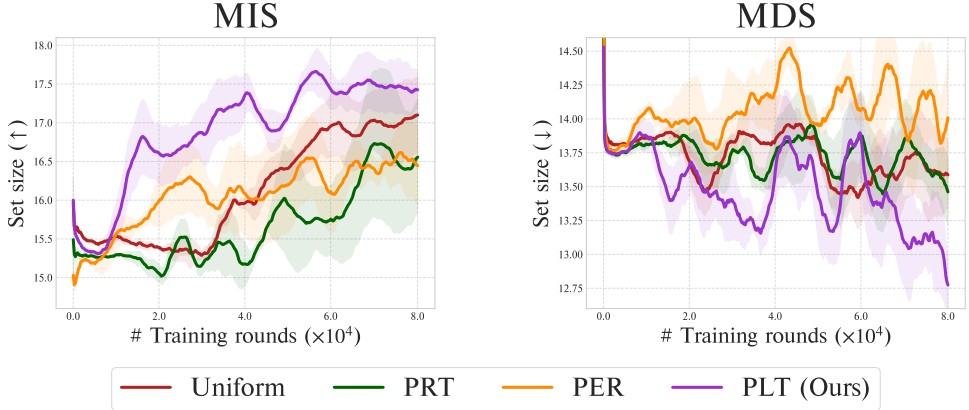

*Figure 6.* Main experiment results in combinatorial optimization tasks. We report the metric size over the course of training. Experiments are conducted with three different seeds.

**L14-RNA**: Generate RNA sequence with 14 nucleotides that maximizes a binding affinity to a human transcription factor, which is obtained via a pre-trained proxy model from Sinai et al. (2020).

**Evaluation Results.** We summarize the number of discovered modes over the course of training in Figure 5. As illustrated in the figure, our method outperforms other off-policy techniques. While PRT exhibits a relatively high performance, it occasionally converges to suboptimal regions, as illustrated by the learning curve of the QM9 task. Notably, as the search space becomes larger (e.g., sEH and L14-RNA), the margin between our method and other baselines increases, indicating the practicality of our method in off-policy training of GFlowNets. We also report the top-$k$ performance of generated samples in Section B.2.

### 4.3. Combinatorial Optimization

Lastly, we conduct experiments on combinatorial optimization tasks, where GFlowNets is a powerful alternative to classical methods due to its capability of handling highly structured constraints. Following Zhang et al. (2023), we conduct experiments on two graph combinatorial optimiza-

tion benchmarks: MIS and MDS. Please refer to Section A.3 for more details of the experimental setup.

**MIS.** Our objective is to discover a maximum independent set of vertices for a given graph, which can be represented as a binary vector, $\mathbf{x} = (x^1, \cdots, x^{|V|})$, where $x^i = 1$ indicates that the $i$-th vertex belongs to the set and $x^i = 0$ indicates that it does not. The agent starts with the void initial state $s_0 = (\emptyset, \cdots, \emptyset)$, and the action is to choose an unspecified vertex and include it in the current set. After performing an action, we identify the vertices that fail to meet this requirement and mark them as $0$. The reward is given as the size of the discovered set.

**MDS.** Similar to MIS, our objective is to discover a minimum dominating set of vertices for a given graph. We start with the unvoid initial state $s_0 = (1, \cdots, 1)$, and the action corresponds to selecting a vertex and removing it from the current set. The reward is given as a minus of the size of the discovered set.

**Evaluation Results.** For evaluation, we sample 50 solutions for each validation graph to measure the average scores at each training step. The results are summarized in Figure 6. As depicted in the figure, our method outperforms

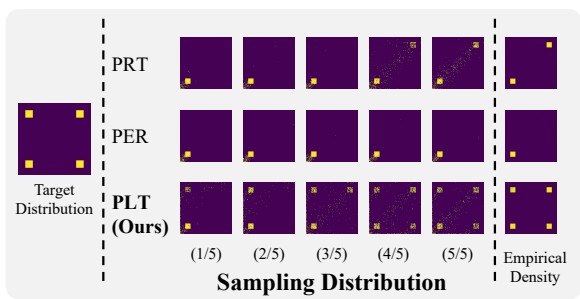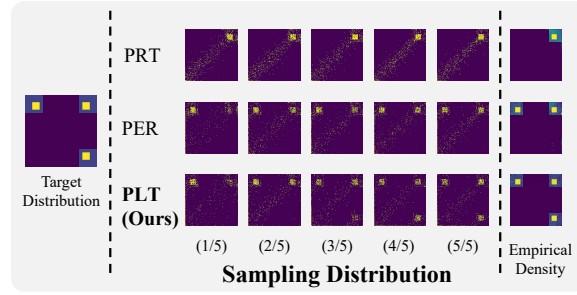

*Figure 7.* Visualization of sampling distributions from the replay buffer in sparse and asymmetric grids.

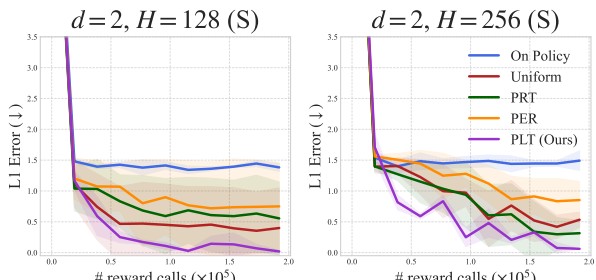

*Figure 8.* Log partition function estimation error on gridworld environments. Experiments are conducted with five different seeds.

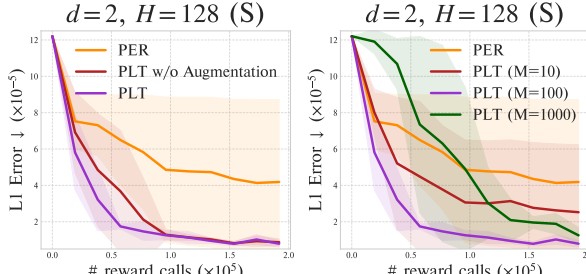

*Figure 9.* Ablation studies on reward mixing term and $M$. Experiments are conducted with five different seeds.

other baselines, demonstrating its versatility. Moreover, we discover that PER performs worse in both tasks, indicating that naively relying on loss values makes learning unstable. We also summarize the results of top-k scoring samples for each method in Section B.3.

## 5. Additional Analysis

### 5.1. Visualization of sampling distributions

We first visualize the sampling distributions of several off-policy methods, including PLT, on the gridworld environment in Figure 7. As illustrated in the figure, our method progressively explores diverse modes as training goes by, whereas other techniques, such as PRT and PER, occasionally become stuck in only a few modes and cannot be further explored. It clearly supports the claim that incorporating learnability when choosing samples from the replay buffer is crucial for the efficacy of GFlowNet training.

### 5.2. Estimation error of log partition function

To analyze the learning stability of PLT, we visualize the absolute error on the log partition function ($\log Z$) across training. As shown in Figure 8, PLT clearly outperforms other baselines in terms of the partition function estimation error, indicating the superiority in terms of learning stability.

### 5.3. Ablation Studies

In this section, we conduct ablation studies of several decision choices in our method, specifically the reward mixing and the reference policy update interval $M$.

**Effect of reward mixing.** To promote exploration towards both high learnability and high-rewarding regions, we augment the reward term by introducing a mixing coefficient $\alpha$. To demonstrate the effectiveness of mixing, we remove the reward augmentation term by setting $\alpha = 0$ and compare the results. As shown in Figure 9, we find that the augmented reward term leads to more efficient training of GFlowNets. Nevertheless, we discover that even with focusing on learnability, we achieve a high efficacy compared to just relying on loss values, demonstrating the importance of learnability-based sampling. We also conduct a sensitivity analysis of $\alpha$ in Section B.4 and observe that there is no big difference when we use different values of $\alpha$.

**Effect of update interval $M$.** To identify learnable states with the current policy, we introduce a reference policy, a lagged version of the current policy updated every $M$ training steps. While we used a fixed value of $M = 100$ across all experiments conducted in Section 4, we evaluate our method by varying $M$ to demonstrate the effect of $M$. As shown in Figure 9, we find that updating a reference policy too often leads to suboptimal results. While increasing $M$ eventually converges to a low L1 density error, it suffers from over-exploration at early stages, similar to PER. We also conduct sensitivity analysis on $M$ in other benchmarks and summarize the results in Section B.5.

### 5.4. Robustness on other GFlowNets variants

In this section, we show that our method can be implemented as a plug-and-play module for various GFlowNet algorithms, including advanced loss functions and exploratory policies.

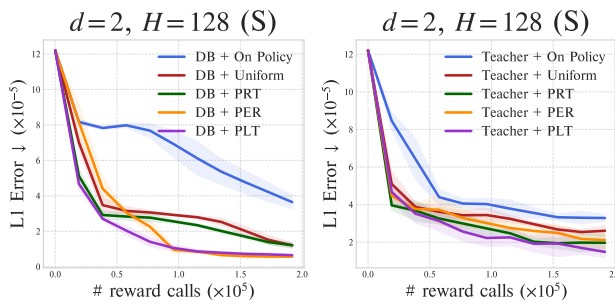

*Figure 10.* Robustness on different objectives and exploratory strategies. Experiments are conducted with five different seeds.

**Robustness on different GFlowNets objectives.** In gridworld, we use TB as a default objective across different off-policy techniques. To verify the robustness of our method with respect to loss functions, we conduct experiments on the gridworld environment using the DB objective. As depicted in Figure 10, our method consistently outperforms other baselines even with a change of loss functions. In addition, we evaluate other objective functions across various benchmarks and provide a summary in Section B.6.

**Robustness on different exploratory strategies.** In gridworld, we use the current policy as a behavior policy for searching candidates. To verify the robustness of our method with respect to exploratory strategies, we conduct experiments on the gridworld environment with Teacher (Kim et al., 2025c) as an exploratory policy. As presented in Figure 10, our method consistently outperforms other baselines when attached with the Teacher policy. Additional results with different objective functions across various benchmarks are provided in Section B.7.

**Robustness on update-to-data ratio.** In off-policy training, the update-to-data (UTD) ratio is crucial for the sample efficiency of the agent (D'Oro et al., 2023). We also conduct experiments by varying the UTD ratio and summarize the results in Section B.8. We observe that as the UTD ratio increases, the gap between PLT and other baselines becomes larger, demonstrating its superiority in off-policy training.

## 6. Related Works

**GFlowNets and following works.** GFlowNets were first introduced by Bengio et al. (2021) to generate non-iterative diverse candidates in molecular graphs and generally extended by Bengio et al. (2023). Bengio et al. (2021) introduced flow matching (FM) objective to learn a stochastic policy that samples proportional to its reward, inspired by temporal difference learning in RL (Sutton et al., 1998). Following FM, several methods have been discussed to improve GFlowNets training. Malkin et al. (2022) introduced the TB objective for better credit assignment over long trajectories by directly learning the partition function, and Madan et al.

(2023) introduced SubTB, which computes losses with sub-trajectories, trading off bias and variance based on TD($\lambda$) principle in RL (Sutton et al., 1998). Shen et al. (2023) and Jang et al. (2024) proposed an improved objective for learning backward policies. Finally, Pan et al. (2024) introduced intrinsic rewards in GFlowNet training to discover diverse modes in a sparse reward landscape.

Orthogonal to the works mentioned above, several methods have been studied to improve off-policy training of GFlowNets. Kim et al. (2024) introduced local search to guide GFlowNets towards high-rewarding regions, while Pan et al. (2024) augments an intrinsic reward term based on novelty to promote exploration in a sparse reward landscape. Recently, several methods have been proposed to train an auxiliary GFlowNet policy to guide the behavior policy exploration towards missing modes and underexplored regions (Kim et al., 2025c; Madan et al., 2025; Malek et al., 2025). In this work, we focus on an underexplored area of research: the sampling distribution from the replay buffer during off-policy training of GFlowNets.

**Adaptive sampling distributions.** Our work is in line with research on adapting sampling distributions during training. In curriculum learning (Bengio et al., 2009), we appropriately schedule the difficulty of tasks from easy to hard, improving the generalization capability of the agent. In active learning (Gal et al., 2017), we actively select samples that yield the maximum information gain to minimize training costs. The key difference is that our method is designed for GFlowNet training, where we need to cover multiple distinct modes of the reward landscape.

## 7. Conclusion

We introduce PLT, a novel sampling distribution for choosing samples from the replay buffer during off-policy training of GFlowNets. PLT promotes selecting both highly learnable and highly rewarding samples, which naturally improve the efficacy of GFlowNet policy in a large state space and sparse reward distribution. We validate the effectiveness of our method across diverse benchmarks, ranging from synthetic grids to combinatorial optimization tasks. Finally, we demonstrate that our method can be implemented as a plug-and-play module on top of diverse GFlowNets objectives and exploratory strategies.

**Limitation and Future works** Our proposed method has numerous possible future research directions. While we focus on discrete objects, which are mostly concerned in GFlowNets, we can extend our method into continuous domains such as diffusion samplers (Sendera et al., 2024) or amortized posterior inference under a generative model prior (Venkatraman et al., 2024; 2025).

## Impact Statement

This work aims to develop better sampling distribution during the off-policy training of GFlowNets. While the proposed approach may impact scientific discovery and combinatorial optimization applications, we believe that the technique does not introduce new application-specific risks, and we do not foresee immediate negative societal or ethical consequences.

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

# Appendix

# A. Task and Implementation Details

### A.1. Gridworld

For the gridworld environment, we strictly follow the implementation of (Bengio et al., 2021). To evaluate the performance, we generate $10^5$ samples with the trained policy $P_F$ and compute empirical L1 distance by $\frac{1}{|\mathcal{X}|} \sum_{x \in \mathcal{X}} |p_\theta(x) - R(x)/Z|$, $Z = \sum_{x \in \mathcal{X}} R(x)$. To visualize the sampling distributions in Figure 7, we store the $2,500$ recently sampled terminal states from the buffer for each off-policy training method, and duplicate samples are not visually distinguished. To parametrize the policy $P_F(\cdot; \theta)$, we use two-layer MLP with 256 hidden units along with the learnable parameter for $\log Z_\theta$. The backward policy $P_B$ is fixed as a uniform policy. For all off-policy methods, we use $\epsilon$-greedy policy as an exploratory policy with $\epsilon = 0.01$ and set update-to-data (UTD) ratio as 4. For implementation of PLT, we use a mixing coefficient $\alpha = 10.0$ for $d = 2$ and $\alpha = 5.0$ for $d = 3, 4$. We fix the update interval $M$ as 100. We summarize the hyperparameters for training in Table 2.

Table 2. Hyperparameters for Gridworld Task

|  | Parameters | Values |
|---|---|---|
| Architecture | Number of Layers | 2 |
|  | Num Units | 256 |
| Training | Batch size | 16 |
|  | Optimizer | Adam |
|  | Learning Rate | $1 \times 10^{-3}$ |
|  | Training Steps | $12,000$ |
|  | UTD ratio | 4 |
|  | Update interval $M$ | 100 |
|  | Mixing Coefficient $\alpha$ | 10.0 ($d = 2$) / 5.0 ($d = 3, 4$) |

### A.2. Biological and Chemical Discovery

For the biochemical discovery, we strictly follow the implementation of (Kim et al., 2024). To evaluate the performance, we define *modes*, high-rewarding samples that are distinct from other samples based on a pre-defined similarity constraint. For QM9 and TFBind8, we use a default set of modes suggested in (Shen et al., 2023). For sEH, we set the reward threshold as top $0.01\%$ of $\mathcal{X}$ in terms of the reward and diversity threshold as $0.4$ Tanimoto diversity. For L14-RNA, we set the reward threshold as top $0.01\%$ of $\mathcal{X}$ in terms of the reward and diversity threshold as 1 Levenstein diversity. For the reward exponent term ($\beta$), we use a higher one to measure the ability of various off-policy methods in sparse reward landscape. We use $\beta = 5$ for QM9, $\beta = 10$ for sEH and TFBind8, and $\beta = 15$ for L14-RNA task. To parametrize the policy $P_F(\cdot; \theta)$, we use two-layer MLP with 128 hidden units for biological sequence design tasks and 1024 molecular graph generation tasks along with the learnable parameter for $\log Z_\theta$. The backward policy $P_B$ is fixed as a uniform policy. For implementation of PLT, we use a mixing coefficient $\alpha = 10.0$ for QM9 and TFBind8, and $\alpha = 500.0$ for sEH and L14-RNA. We fix the update interval $M$ as 100. We summarize the hyperparameters for training in Table 3.

Table 3. Hyperparameters for Biochemical Discovery

|  | Parameters | Values |
|---|---|---|
| Architecture | Number of Layers | 2 |
|  | Num Units | 1024 (QM9, sEH) / 128 (TFBind8, L14-RNA) |
| Training | Batch size | 32 |
|  | Optimizer | Adam |
|  | Learning Rate | $1 \times 10^{-2}$ ($\log Z_\theta$) / $1 \times 10^{-4}$ ($P_F(\cdot; \theta)$) |
|  | Training Steps | $2,000$ |
|  | Update interval $M$ | 100 |
|  | Mixing Coefficient $\alpha$ | 10.0 (QM9, TFBind8) / 500.0 (sEH, L14-RNA) |

### A.3. Combinatorial Optimization

For the combinatorial optimization, we strictly follow the implementation of (Zhang et al., 2023). For both MIS and MDS tasks, we sample $4,000$ graphs with $|\mathcal{V}| \sim \text{Unif}(200, 300)$ for training and $400$ graphs for evaluation. To evaluate the performance, we sample $50$ solutions for each validation graph to measure the average scores.

In combinatorial optimization tasks, we use the DB objective, which is the default objective. To parametrize the policy $P_F(\cdot; \theta)$ and the flow function $F(\cdot; \theta)$, we use a graph isomorphism network (?)GIN,][]xu2018powerful with five hidden layers and a 256-dimensional hidden size. For training, we sample $64$ transitions from the replay buffer for each training step. As the default strategy is sampling transitions from the buffer, we bypass the on-policy baseline. Following (Zhang et al., 2023), we reset the buffer for each training epoch. For the implementation of PLT, we use a mixing coefficient $\alpha = 10.0$ and the update interval $M$ as 100 across all experiments. We summarize the hyperparameters for training in Table 4.

*Table 4.* Hyperparameters for Combinatorial Optimization

|  | Parameters | Values |
|---|---|---|
| Architecture | Number of Layers | 5 |
|  | Num Units | 256 |
| Training | Batch size | 64 |
|  | Optimizer | Adam |
|  | Learning Rate | $1 \times 10^{-3}$ |
|  | Training Epochs | 20 |
|  | Update interval $M$ | 100 |
|  | Mixing Coefficient $\alpha$ | 10.0 |

### A.4. Computational Resources and Runtime

We only use the CPU for running gridworld experiments. For running biochemical tasks, we use a single RTX 3090 GPU. For combinatorial optimization tasks, we use a single L40S GPU. We summarize the runtime of each method in Table 5. As shown in the table, there are no big differences between off-policy training methods in terms of runtime. It is because PLT only requires a single additional forward pass with the reference policy.

*Table 5.* Runtime of each method across different benchmarks. Minutes are truncated in the table.

| Method | Gridworld | | | | Biochemical Discovery | | | | Combinatorial Optimization | |
|---|---|---|---|---|---|---|---|---|---|---|
|  | $d=2, H=128$ | $d=2, H=256$ | $d=3, H=64$ | $d=4, H=48$ | QM9 | sEH | TFBind8 | L14-RNA | MIS | MDS |
| On-Policy | 1h 30m | 2h 40m | 1h 40m | 2h 40m | 30m | 4h 40m | 1h 30m | 3h | - | - |
| Uniform | 2h 30m | 4h 30m | 2h 30m | 3h 30m | 40m | 5h 30m | 1h 40m | 5h 30m | 4h | 1d 2h |
| PRT | 2h 40m | 4h 40m | 2h 40m | 3h 40m | 40m | 5h 30m | 1h 40m | 5h 30m | 4h | 1d 2h |
| PER | 2h 40m | 4h 40m | 2h 30m | 3h 40m | 40m | 5h 30m | 1h 40m | 5h 30m | 4h 40m | 1d 3h |
| PLT (Ours) | 2h 40m | 4h 50m | 2h 40m | 3h 40m | 40m | 5h 30m | 1h 40m | 5h 30m | 4h 40m | 1d 4h |

# B. Extended Additional Analysis

In this section, we provide further analysis on different components of our method that are not included in the main manuscript due to the page limit.

## B.1. Number of Discovered Modes in Gridworld

In Figure 4, we visualize the empirical L1 density error over the course of training. In Figure 11, we visualize the number of discovered modes across training to demonstrate the mode coverage capability of our method. For the sparse grid, we define modes as a cell with a reward $R_0 + R_2$, and for the asymmetric grid world, we define modes as a cell with a reward $R_0 + R_1 + R_2$. As shown in the figure, PLT clearly outperforms other baselines in terms of the number of discovered modes, indicating the superiority in terms of mode coverage.

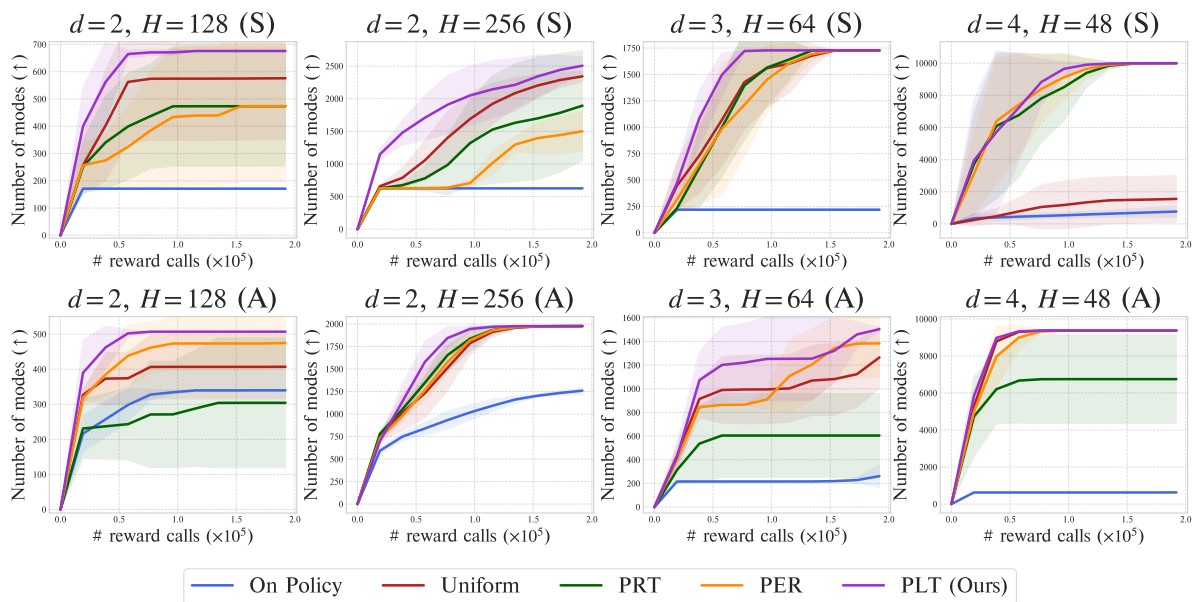

*Figure 11.* Number of discovered modes over the course of training in gridworld environments. Experiments are conducted with five different seeds.

## B.2. Top-K Performance in Biological and Chemical Discovery

In Figure 5, we visualize the number of discovered modes over the course of training. In Figure 12, we visualize the top-k performance of generated samples from the trained policy across training. For every 100 training steps, we generate 2,048 samples from the policy and report the average reward of top-128 samples. As shown in the figure, PLT outperforms other baselines in terms of top-k performance, demonstrating its effectiveness.

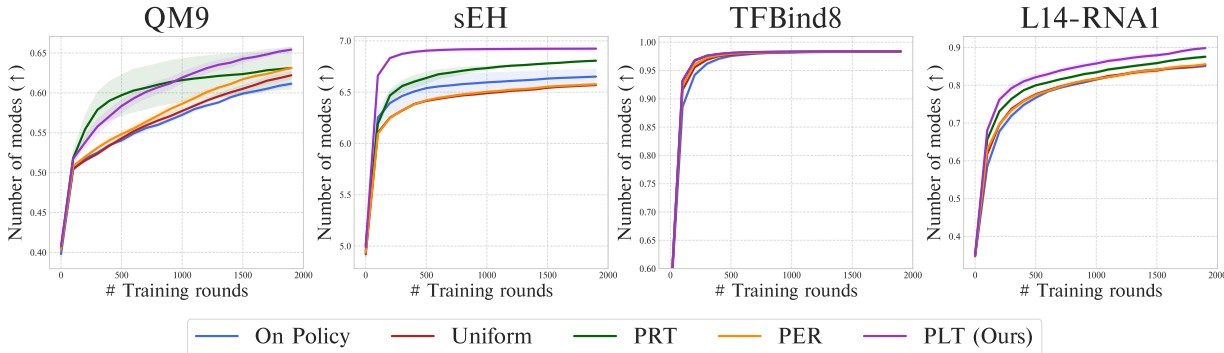

*Figure 12.* We report the mean score of top-128 samples generated by each method on four biochemical tasks. Experiments are conducted with four different seeds.

## B.3. Top-K Performance in Combinatorial Optimization

In Figure 5, we visualize the performance of different off-policy methods by sampling 50 solutions for each validation graph and measuring the average scores. In Figure 13, we visualize the top-k performance of generated solutions from the trained policy across training. For every epoch, we sample 50 solutions for each validation graph and report the average set size of top-20 solutions. As shown in the figure, PLT outperforms other baselines in terms of top-k performance, demonstrating its effectiveness.

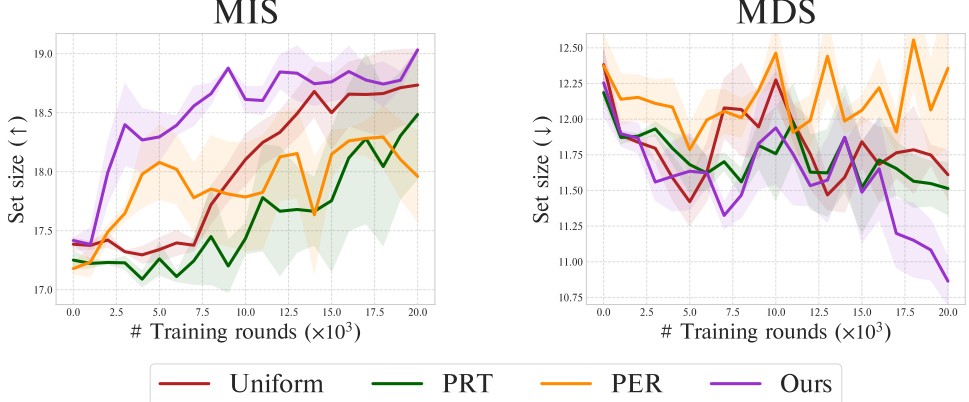

*Figure 13.* We report the mean score of top-20 solutions generated by each method on two combinatorial optimization tasks. Experiments are conducted with three different seeds.

## B.4. Sensitivity Analysis on $\alpha$

In Figure 9, we conduct an analysis on the reward augmentation term and confirm that while the sample efficiency of PLT without reward augmentation slightly drops, it still outperforms PER, demonstrating the importance of learnability information. In Figure 14, we conduct an analysis on the choice of mixing coefficient $\alpha$ by running experiments with different $\alpha$ values. As shown in the figure, there is no big difference when we use different $\alpha$, demonstrating its robustness. Note that we fix $\alpha = 10.0$ for main gridworld experiments with $d = 2$.

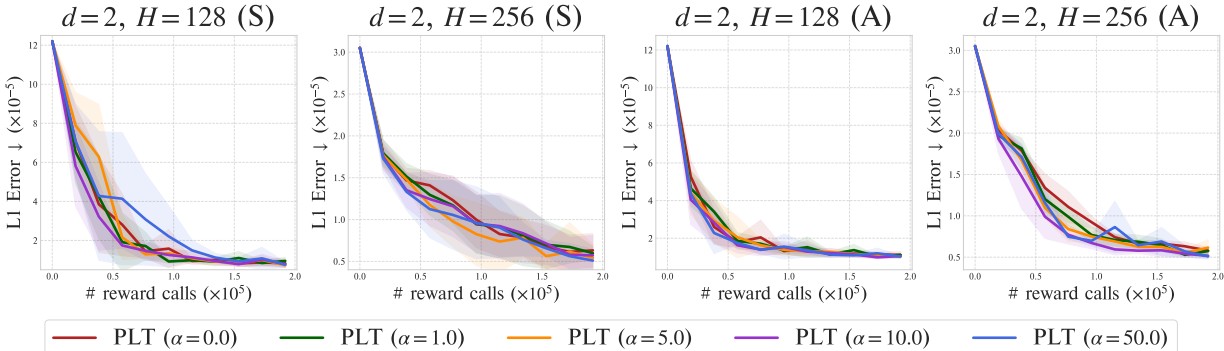

*Figure 14.* Sensitivity analysis on $\alpha$ in gridworld environments. Experiments are conducted with five different seeds.

## B.5. Sensitivity Analysis on $M$

In Figure 9, we conduct an analysis on reference policy update interval $M$ on $d = 2, H = 128$ sparse grid. In Figure 15, we visualize the results of other gridworld environments and confirm that we get a consistent result: Increasing $M$ exhibits a slow learning curve, while decreasing $M$ can suffer from learning instability. Note that we fix $M = 100$ for all main experiments.

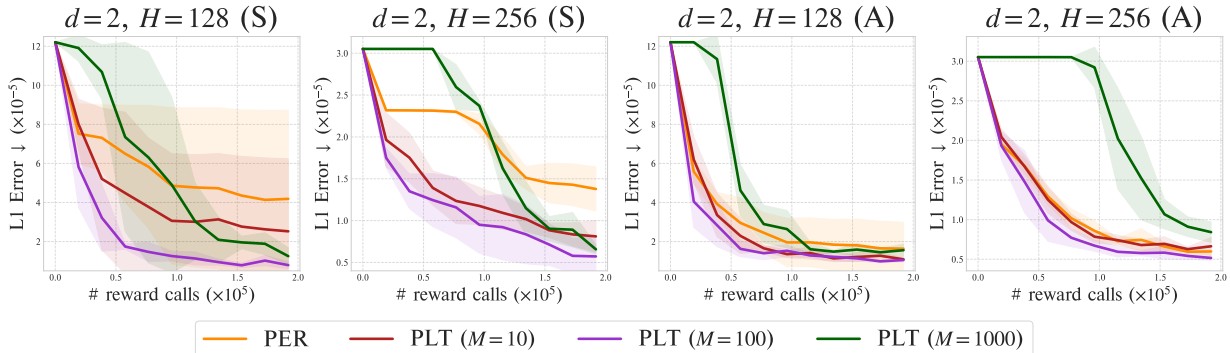

*Figure 15.* Sensitivity analysis on $M$ in gridworld environments. Experiments are conducted with five different seeds.

## B.6. Robustness on Different GFlowNet Objectives

In Figure 5, we use the TB objective for training various off-policy methods. In Figure 16, we visualize the number of discovered modes during training on two biochemical tasks when we use the DB objective for training various off-policy methods. As shown in the figure, we observe similar trends with experiments using the TB objective. PLT consistently outperforms other baselines with different training objectives, demonstrating its robustness.

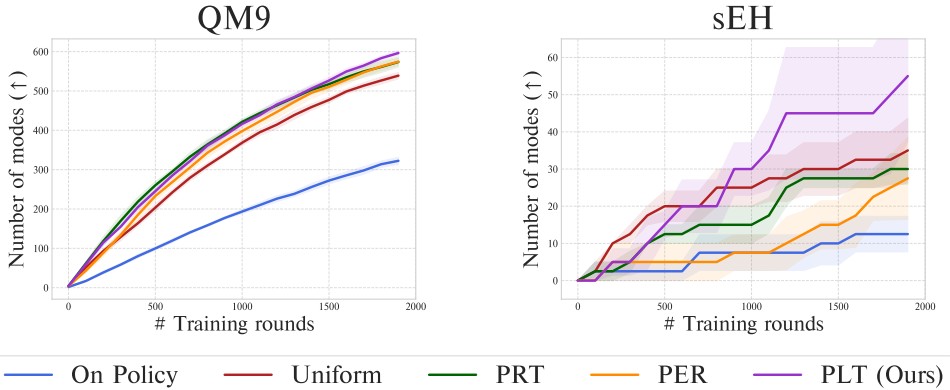

*Figure 16.* We report the number of discovered modes in biochemical tasks with DB as a training objective. Experiments are conducted with four different seeds.

## B.7. Robustness on Different Exploratory Strategies

In Figure 5, we utilize the current policy as an exploratory policy for training various off-policy methods. In Figure 17, we visualize the results when we use the local search suggested by (Kim et al., 2024) as an exploratory policy for training various off-policy methods. As shown in the figure, PRT exhibits high performance in the early rounds but converges to suboptimal solutions due to its vulnerability to mode collapse. In contrast, PLT consistently outperforms other baselines with different exploratory policies, demonstrating its robustness.

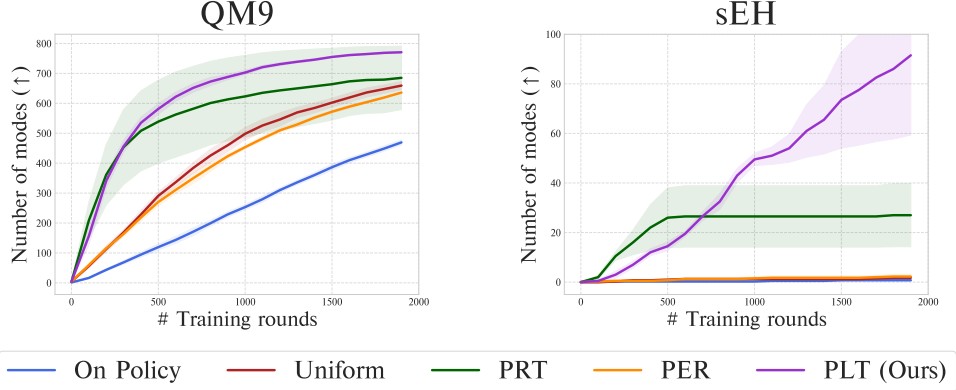

*Figure 17.* We report the number of discovered modes in biochemical tasks with local search as an exploratory policy. Experiments are conducted with four different seeds.

### B.8. Robustness on Update-to-data (UTD) ratio of Off-policy Training

In off-policy training, the update-to-data (UTD) ratio, which indicates the number of gradient steps per on-policy sampling, can be crucial in the sample-efficient training. To this end, we conduct experiments by varying the UTD ratio from 1 to 8. As depicted in Figure 18, PLT shows consistent improvement compared to different off-policy techniques across different UTD ratios. Moreover, we observe that as the UTD ratio increases, the gap between PLT and other baselines becomes larger, demonstrating its superiority in off-policy training.

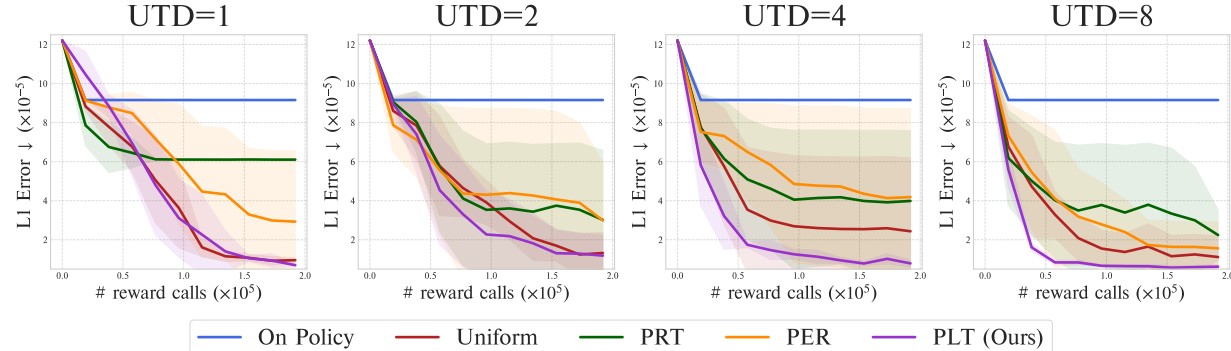

*Figure 18.* Robustness on update-to-data (UTD) ratio of off-policy training in gridworld environments. Experiments are conducted with five different seeds.

