# OpenReview forum: "Improving Sampling Distribution of Off-policy Training in Generative Flow Networks"
_ICML.cc/2026/Conference — Submitted to ICML 2026_

### Official Review · Reviewer_9bXh · 2026-02-20

**Soundness:** 3
**Presentation:** 1
**Significance:** 3
**Originality:** 2
**Overall Recommendation:** 4
**Confidence:** 3

**Summary:**

This paper introduces a novel **off-policy sampling strategy for Generative Flow Networks (GFlowNets)**, which diverges from previous methods by leveraging a heuristic based on the loss difference between the current policy and a lagged version of the previous policy. The method is evaluated across diverse benchmarks, including **Gridworld**, **biological and chemical discovery problems**, and **combinatorial optimization tasks** (e.g., Maximum Independent Set (MIS) and Minimum Dominating Set (MDS)). Comparative results demonstrate that the proposed strategy outperforms baselines such as **on-policy GFlowNets, uniform sampling, Prioritized Experience Replay (PER), and prioritized replay training (PRT)**.

**Compliance With Llm Reviewing Policy:**

Affirmed.

**Final Justification:**

The paper appears technically sound, and the authors show that their method outperforms existing off-policy sampling strategies.
I initially had concerns about the experiments on the combinatorial optimization dataset, as this part was not explained clearly enough in the original draft. However, the authors addressed these concerns in their response and added additional baselines, which improved my confidence in the experimental evaluation.
The main weaknesses are therefore mostly related to presentation. In particular, some parts of the paper suffered from insufficient explanation, broken references, and inconsistent notation. Another weakness is that the core idea is relatively simple. That said, since I believe the paper is technically correct, I assign only limited negative weight to this point.
On the positive side, the method is shown to perform well across a sufficient number of benchmarks. Its simplicity can also be seen as a practical advantage, especially in terms of ease of implementation.

**Key Questions For Authors:**

## Questions and Suggestions
1. **Comparison with Prior Off-Policy Schemes**
   Results show that previous off-policy sampling schemes (e.g., PER, PLT) do not outperform uniform sampling. Is this observation consistent with findings from the original PER and PLT papers? Clarifying this would strengthen the motivation for the proposed method.

2. **Evaluation on Harder Tasks**
   - To better demonstrate the method’s capabilities, I recommend further evaluation on more challenging MIS instances, such as those generated by the RB model [2, 3].
   - Including baselines from other methods beyond GflowNets would provide a clearer picture of the benchmarks’ difficulty and the method’s relative performance.

---

## References
[1] Xu, Ke, et al. "A simple model to generate hard satisfiable instances." *arXiv preprint cs/0509032* (2005).
[2] Wang, Haoyu, and Pan Li. "Unsupervised learning for combinatorial optimization needs meta-learning." *arXiv preprint arXiv:2301.03116* (2023).
[3] Sanokowski, Sebastian, Sepp Hochreiter, and Sebastian Lehner. "A diffusion model framework for unsupervised neural combinatorial optimization." *arXiv preprint arXiv:2406.01661* (2024).

**Limitations:**

Yes

**Strengths And Weaknesses:**

## Strengths
1. **Innovative Research Direction**
   The exploration of improved off-policy sampling strategies is both timely and relevant, addressing a critical gap in the field.

2. **Empirical Performance**
   The proposed method consistently outperforms existing approaches, as evidenced by the experimental results.

---

## Weaknesses
1. **Presentation Issues**
   - **Broken References:** Several references are incomplete or missing (e.g., lines 611 and 267), marked as `??` or `?`, which disrupts readability and credibility.
   - **Inconsistent Notation:** Equation (7) and other formulas exhibit inconsistencies, which may confuse readers and undermine the paper’s rigor.

2. **Lack of Experimental Clarity**
   - **Combinatorial Optimization (CO) Benchmarks:** While the appendix describes how the number of vertices is sampled, critical details such as graph connectivity and graph type remain unspecified. To ensure robustness, it is essential to evaluate the method on *hard* CO instances, such as those generated by the **RB model** [1], where prior works have assessed performance across varying hardness levels [2, 3].
   - **Limited Comparative Analysis:** The paper does not benchmark against methods beyond GFlowNets, making it difficult to contextualize the difficulty of the problems and the relative performance of other approaches.
3. Benchmarks seem to be rather toyish

---

> ### Author Rebuttal · Authors · 2026-03-31
>
> We sincerely thank the reviewer for the valuable feedback and acknowledging novelty and extensive empirical results of our method. We address your specific concerns as below:
>
> > **(W1)** Presentation Issues
>
> We appreciate the reviewer's careful reading and sincerely apologize for these formatting issues. We will fix all broken references and typos in the revised version:
>
> - References: The ?? in line 267 refers to Section 5.2 (regarding the estimation error of log partition function). The broken citation in line 611 refers to GIN [1].
> - Notations: We have updated the notation in Eq. (7) ($\text{Learnability}(\tau_{\rightarrow x})=| \ell(\tau_{\to x}; \theta) - \ell(\tau_{\to x}; \hat{\theta})|$). For Algorithm 1, we abbreviate the notation of $\tau_{\rightarrow x_i}$ as $\tau_i$ due to the space limit. We will thoroughly proofread and revise all related notations in the manuscript.
>
> [1] Xu, Keyulu, et al. "How Powerful are Graph Neural Networks?." ICLR 2019.
>
> > **(W2 & Q2-1)** Lack of Experimental Clarity / Evaluation on Harder Tasks
>
> We would like to clarify that the training and validation graphs utilized in our CO experiments were indeed already generated using the RB model as the reviewer suggested [1, 3]. Specifically, we sampled the number of vertices uniformly from 200 to 300 to generate the graph instances. For the dataset split, we used 4,000 graphs for training, and 500 graphs each for the validation and test sets.
>
> To further address the reviewer's concern about the robustness of our method on highly challenging CO instances, we have conducted an additional evaluation on a much larger scale. Following the methodology in the mentioned paper [3], we significantly increased the size of the RB-generated graphs, scaling from the original $|V|\sim U[200,300]$ up to $|V|\sim U[800,1200]$. As shown in the table below, PLT seamlessly scales to these larger instances and maintains a consistent performance improvement over the baselines.
>
> **Experiment results on combinatorial optimization tasks with larger scale.**
> |    | MIS ($\vert V\vert \sim U[800, 1200]$) |
> | ---| --- |
> | Uniform    | 37.50 ± 1.11 |
> | PRT        | 36.39 ± 0.59 |
> | PER        | 38.24 ± 0.17 |
> | PLT (ours) | **38.51 ± 0.26**|
>
> > **(W3)** Benchmarks seem to be rather toyish
>
> We would like to clarify that our original experiments follow the standard evaluation protocols widely adopted in the GFlowNet literature, which span multiple domains, including combinatorial optimization tasks that feature extremely large search spaces. Nevertheless, we appreciate the reviewer's feedback and conduct additional experiments with the GFP task, a more challenging benchmark as suggested by [4] (the search space is combinatorially large, $\vert\mathcal{X}\vert=20^{238}$). As shown in the table, we found that augmenting TB with PLT yields novel, high-quality candidates compared to the original baseline.
>
> **Experiment results on GFP task. Experiments are conducted with four random seeds.**
> |          | Top-K performance | Top-K Novelty |
> |:-------- |:---| :--- |
> | GFN-AL (TB)       |3.573 ± 0.003|10.777 ± 1.806|
> | GFN-AL (TB + PLT) |**3.577 ± 0.002**|**13.257 ± 0.325**|
>
> [4] Jain, Moksh, et al. "Biological sequence design with gflownets." ICML 2022.
>
>
> > **(Q1)** Comparison with Prior Off-Policy Schemes
>
> We thank the reviewer for the suggestion. In simple Gridworld environments with dense rewards, PRT and PER typically outperform uniform sampling. However, we deliberately tailor the environments closer to realistic settings, where reward is highly sparse and asymmetric. In such challenging environments, traditional prioritization schemes suffer from severe mode collapse and premature exploitation.
> Unlike prior baselines, our method does not suffer from mode collapse or high variance issues due to the effectiveness of the learnability term. We will include the clarification in the revised manuscript.
>
> >  **(Q2-2)** Stronger Baselines
>
> We thank the reviewer for this constructive suggestion.
> To address this concern, we include two additional strong alternative methodologies for CO tasks: PPO [5] (an RL-based approach) and DGL (a supervised learning approach with tree search refinement) [6].
>
> As shown in the newly added table below, PLT not only improves upon prior off-policy GFlowNet sampling methods but also consistently outperforms the baselines. It confirms that reward-matching GFlowNets equipped with our learnability-based sampling distribution yield strong performance on these challenging tasks.
>
> **Experiment results on combinatorial optimization tasks compared to baselines beyond GFlowNets.**
> |     | MIS ($\vert V\vert \sim U[800, 1200]$) |
> | --- | --- |
> | PPO | 32.32|
> | DGL | 34.50|
> | GFlowNets | 37.50 ± 1.11 |
> | GFlowNets + PLT (ours) | **38.51 ± 0.26**|
>
> [5] Ahn et al., Learning what to defer for maximum independent sets. ICML 2020.
>
> [6] Böther et al., What’s wrong with deep learning in tree search for combinatorial optimization. ICLR 2022.

---

> > ### Author Rebuttal · Reviewer_9bXh · 2026-04-01
> >
> > We appreciate the authors’ response to our comments.
> >
> > However, I still have concerns regarding the **CO experiments conducted on the RB-model**. While the authors provide the node ranges, generating RB model graph instances requires specifying several critical hyperparameters:
> > - The number of cliques,
> > - The number of members within each clique, and
> > - The parameter **p**, which determines the interconnectivity of the cliques.
> >
> > The value of **p** is particularly important, as it directly influences the hardness of the problem (e.g., a **p** value of 1 renders the problem trivial). Reporting these values is essential not only for reproducibility but also for assessing the problem’s difficulty.
> >
> > Additionally, we kindly request that the authors include results for **DB-Greedy** [1] as a baseline. This method is straightforward to implement and serves as a useful indicator of problem hardness. Furthermore, it would be valuable to report the **optimal Maximum Independent Set (MIS) value** achievable on this dataset.
> >
> > [1] Toenshoff, Jan, et al. "Graph neural networks for maximum constraint satisfaction." *Frontiers in Artificial Intelligence* 3 (2021): 580607.

---

> > > ### Author Response · Authors · 2026-04-01
> > >
> > > We thank the reviewer for the continued engagement and valuable suggestions. We address your additional questions below:
> > >
> > > Regarding the RB-model generation, we strictly followed the standard protocol established in [3]. The critical hyperparameters are sampled as follows:
> > > - The number of cliques $n\in [40, 55]$
> > > - The number of members within each clique $k\in [20, 25]$ (graphs that are smaller than 800 nodes or larger than 1200 nodes are resampled)
> > > - The parameter $p$ $\in [0.3, 1.0]$ (guarantees that the generated MIS instances are non-trivial in general)
> > >
> > > We will explicitly include these parameters and the justification for problem hardness in the revision.
> > >
> > > Following your constructive suggestion, we have evaluated DB-Greedy to serve as a practical indicator of problem hardness. Furthermore, we calculated the optimal Maximum Independent Set (MIS) values using the Gurobi solver.
> > >
> > > For a fair comparison, the DB-Greedy baseline was trained using the same number of epochs as GFlowNet methods. For the Gurobi solver, we allocated a maximum evaluation budget of 100 seconds per instance.
> > >
> > > As demonstrated in the table below, our proposed method (GFlowNets + PLT) outperforms the DB-Greedy baseline by a large margin. This suboptimal performance of a straightforward greedy approach also empirically validates that our RB-model dataset is highly challenging. Moreover, we achieves results that are close to the Gurobi, which takes a significantly longer runtime. We will incorporate these results into the revised manuscript.
> > >
> > > Experiment results on combinatorial optimization tasks compared to baselines beyond GFlowNets.
> > > |     | MIS ($\vert V\vert \sim U[800, 1200]$) |
> > > | --- | --- |
> > > | Gurobi  | 42.76 |
> > > | DB-Greedy|34.30 ± 0.43|
> > > | GFlowNets + PLT (ours) | **38.51 ± 0.26**|
> > >
> > > We are grateful for these follow-up suggestions. We hope these updates fully resolve your concerns and might positively influence your final assessment.

---

### Official Review · Reviewer_ECJj · 2026-03-06

**Soundness:** 3
**Presentation:** 2
**Significance:** 3
**Originality:** 3
**Overall Recommendation:** 4
**Confidence:** 3

**Summary:**

The paper introduces Prioritized Learnability Training (PLT), a novel sampling strategy for the replay buffer during the off-policy training of Generative Flow Networks (GFlowNets). Standard off-policy sampling methods, such as uniform sampling, Prioritized Experience Replay (PER), or Prioritized Replay Training (PRT), often struggle with mode collapse or high variance, especially in environments with sparse rewards and large state spaces.

To address this, the authors propose sampling proportional to a "learnability" metric, defined as the absolute difference in loss between the current policy and a lagged reference policy, augmented by a reward term. The method is evaluated across a diverse set of tasks, including synthetic gridworlds, biochemical discovery (QM9, sEH, TFBind8, L14-RNA), and graph combinatorial optimization (MIS, MDS), demonstrating improved mode discovery and learning stability compared to baseline sampling strategies.

**Compliance With Llm Reviewing Policy:**

Affirmed.

**Final Justification:**

The paper makes a meaningful and original contribution to off-policy GFlowNet training by proposing a simple, practical replay-buffer sampling strategy that is well motivated and supported by strong empirical results across diverse benchmarks. I weighed the paper positively on originality, significance, and practical utility, especially given the robust ablations and consistent improvements in mode discovery and stability. My main concerns were the heuristic nature of the learnability metric, some metric-presentation ambiguity, and several formatting issues. The rebuttal partially addressed these concerns: while it did not provide new theoretical guarantees, it gave reasonable clarification about the off-policy setting and clearly committed to improving the metric explanation and fixing presentation errors.  The rebuttal reinforced my view that this is a technically solid and useful contribution, though still somewhat limited by the lack of deeper theoretical grounding, so I maintain my weak accept recommendation.

**Key Questions For Authors:**

1. **Theoretical Justification:** While the empirical results are compelling, can you provide any theoretical intuition, bounds, or convergence analysis to justify why the specific learnability formulation ($|l(\tau_{\rightarrow x};\theta) - l(\tau_{\rightarrow x;}\hat{\theta})| + \alpha \cdot R(x)$) is mathematically sound compared to other potential distance metrics between the two policies?

&nbsp;

2. **Table 1 Metric Clarification:** In Table 1, the reported L1 errors for the Uniform sampler (and others) are numerically smaller on the much larger $H=256$ grid compared to the $H=128$ grid. I assume this is a mathematical artifact of the empirical L1 distance formula dividing by the total number of terminal states $|\mathcal{X}|$. Could you explicitly clarify this scaling effect in the main text or caption so readers do not misinterpret this as the model performing "better" on the harder task?

&nbsp;

3. **Formatting Updates:** Will you commit to fixing the unresolved LaTeX references (e.g., the `??` in Section 4.1), broken citations (in Appendix A.3) and fix formatting of the formula 10 to make it fit into a single column

**Limitations:**

Yes. The authors have included a brief "Limitation and Future works" paragraph in the conclusion discussing the restriction to discrete objects and potential continuous extensions, as well as an "Impact Statement" addressing societal consequences.

**Strengths And Weaknesses:**

**Strengths**

&nbsp;


- **Clear Narrative & Novelty:** The paper is very well-written, with a clear and logical narrative flow supported by intuitive visual aids (like Figure 2 and Figure 7). It addresses a critical bottleneck in GFlowNets—balancing exploration and exploitation during off-policy training—by introducing a clever and novel synthesis of ideas: using the difference between a current and a lagged policy to define "learnability" for the replay buffer.

&nbsp;



- **Robust Empirical Evaluation:** The empirical evaluation is highly robust and well-designed. The authors test their plug-and-play module across a diverse set of domains (synthetic gridworlds, biochemical discovery tasks like QM9 and sEH, and graph combinatorial optimization like MIS and MDS), proving that the method generalizes beyond toy datasets to standard benchmarks in the field.

&nbsp;



- **Practical Utility:** Thorough ablation studies validate the specific components of the proposed method (like the mixing coefficient and update interval), and runtime analysis shows the method adds minimal computational overhead, offering immediate practical utility.


&nbsp;



**Weaknesses**

&nbsp;


- **Lack of Theoretical Guarantees:** The most significant weakness is that the proposed formulation for learnability—specifically the equation $|l(\tau_{\rightarrow x};\theta) - l(\tau_{\rightarrow x;}\hat{\theta})| + [cite_start]\alpha \cdot R(x)$—is entirely heuristic. The paper lacks a theoretical foundation, mathematical proofs, or convergence guarantees to justify why this specific combination is optimal. While the empirical results are strong, the method currently reads as somewhat ad hoc.

&nbsp;



- **Metric Scaling Clarity:** The scaling of the metric in Table 1 can be highly unintuitive for a reader at first glance. Because the L1 density error divides by the total number of terminal states $|\mathcal{X}|$, the Uniform sampler error appears numerically smaller on the vastly larger $H=256$ task compared to $H=128$. This requires explicit clarification in the text so readers do not misinterpret this artifact as the model performing "better" on a harder task.

&nbsp;



- **Formatting Errors:** There are several minor but noticeable formatting and referencing errors that need to be addressed. For instance, in Section 4.1, there is an unresolved reference: "estimation error of the log partition function $(\log Z)$ in ??". Furthermore, in Appendix A.3, there is a broken citation tag: "(?)GIN,][] xu2018powerful". And formula (10) does not fit in a colums

---

> ### Author Rebuttal · Authors · 2026-03-31
>
> We sincerely thank the reviewer for the valuable feedback and for acknowledging the novelty and practicality of our method. We address your specific concerns as below:
>
> > **(W1 & Q1)** Lack of Theoretical Guarantees /  Theoretical Justification
>
> We acknowledge the reviewer's point that the paper does not present a formal proof of variance reduction or optimization stability. Unfortunately, deriving exact bounds on variance reduction for deep neural networks with non-linear function approximators remains a highly non-trivial open challenge in the field.
>
> Rather than theoretical bounds, our work starts from the empirical observation: prior sampling methods suffer from severe mode collapse or high variance, especially in large search spaces with sparse rewards. Our contribution is to provide a simple but effective solution for this problem. As demonstrated across diverse benchmarks—from synthetic grids to biochemical discovery and combinatorial optimization—PLT consistently achieves better mode coverage compared to prior baselines.
>
> While we do not provide a novel convergence proof for PLT, we clarify that sampling distribution does not affect the unbiasedness of the TB/DB objectives thanks to its off-policyness nature [1,2].
>
> [1] Malkin, Nikolay, et al. "Trajectory balance: Improved credit assignment in gflownets." Advances in Neural Information Processing Systems 35 (2022): 5955-5967.
>
> [2] Bengio, Yoshua, et al. "Gflownet foundations." Journal of Machine Learning Research 24.210 (2023): 1-55.
>
> > **(W2 & Q2)** Metric Scaling Clarity / Table 1 Metric Clarification
>
> We thank the reviewer for pointing out this critical nuance. As you mentioned, the error is averaged over the total number of terminal states so it appears numerically smaller as the tasks become larger. We will update the manuscript to clearly articulate the exact mathematical formulation used to compute this evaluation metric to ensure complete transparency.
>
> > **(W3)** Formatting Errors / Formatting Updates
>
> We sincerely apologize for these formatting issue and thank the reviewer for carefully reading of our manuscript. We commit to resolving all of these issues in the revised version.
>
> Specifically, the ?? in Section 4.1 indicates estimation error of log partition function in Section 5.2. Broken citation indicates GIN [3].
>
>
> [3] Xu, Keyulu, et al. "How Powerful are Graph Neural Networks?." International Conference on Learning Representations.

---

> > ### Author Rebuttal · Reviewer_ECJj · 2026-04-03
> >
> > Thank you for providing further clarification. I appreciate the authors' efforts and will incorporate this rebuttal into my final evaluation

---

### Official Review · Reviewer_MH45 · 2026-03-12

**Soundness:** 3
**Presentation:** 3
**Significance:** 3
**Originality:** 3
**Overall Recommendation:** 4
**Confidence:** 1

**Summary:**

This paper proposes Prioritized Learnability Training (PLT) to improve the sampling distribution of the replay buffer during the off-policy training of Generative Flow Networks (GFlowNets). The authors point out that existing off-policy sampling techniques face the problems of high variance and mode collapse, respectively. To solve these problems, PLT measures the "learnability" of samples by computing the absolute difference in loss between the current policy and a lagged reference policy, combined with an augmented reward term. Through extensive evaluation of PLT on gridworld, biochemical discovery tasks, and combinatorial optimization problems, it is demonstrated that this method achieves consistent improvements in mode discovery and convergence speed compared to baseline methods.

**Compliance With Llm Reviewing Policy:**

Affirmed.

**Final Justification:**

Thanks to the authors for the response and the additional experiments. After reading the rebuttal, I will keep my score unchanged.

**Key Questions For Authors:**

1.How sensitive is the PLT mechanism in the paper to the choice of the backward policy $P_B$?

2.Could you provide theoretical insights or proofs to explain why the absolute loss difference specifically correlates with the optimal information gain or exploration metrics for GFlowNets?

3.The paper mentions in Section 5.3 that updating the reference policy too often or too infrequently (high or low update interval $M$) leads to suboptimal results or performance drops. Is there a principled heuristic method to dynamically set $M$ during training based on gradient variance or loss plateauing?

**Limitations:**

yes

**Strengths And Weaknesses:**

(Strengths)

1.This paper identifies the high variance of PER and the vulnerability of PRT to mode collapse, using this as the motivation for the learnability-based approach, which is a practical and strong contribution.

2.The proposed method is highly practical. It can be easily integrated into existing GFlowNet training algorithms as a plug-and-play module, and it incurs minimal additional computational overhead.

(Weaknesses)

1.Although it appears intuitively reasonable, the paper relies primarily on empirical evidence and lacks formal theoretical proofs or bound derivations.

2.PLT introduces new hyperparameters ($\alpha$ and $M$). While the authors demonstrate the robustness of $\alpha$ in the gridworld environment, they do not show whether tuning these parameters remains straightforward and robust in more complex biochemical discovery or combinatorial optimization tasks.

---

> ### Author Rebuttal · Authors · 2026-03-31
>
> We sincerely thank the reviewer for the valuable feedback and acknowledging practicality of our method as a plug-and-play module. We address your specific concerns as below:
>
> > **(W1)** Although it appears intuitively reasonable, the paper relies primarily on empirical evidence and lacks formal theoretical proofs or bound derivations.
>
> We agree with the reviewer that formal variance bounds are theoretically highly desirable. However, deriving exact bounds on variance reduction for deep neural networks utilizing non-linear function approximators over long horizons remains a highly non-trivial open challenge in the field.
>
> Rather than theoretical bounds, we empirically identify the high variance of PER and the vulnerability of PRT to mode collapse. As you mentioned, our simple but effective strategy to design sampling distribution can be easily incorporated in GFlowNet training algoorithms and achieve improvements.
>
> While we do not provide a novel convergence proof for PLT, we clarify that sampling distribution does not affect the unbiasedness of the TB/DB objectives thanks to its off-policyness nature [1,2].
>
> [1] Malkin, Nikolay, et al. "Trajectory balance: Improved credit assignment in gflownets." Advances in Neural Information Processing Systems 35 (2022): 5955-5967.
>
> [2] Bengio, Yoshua, et al. "Gflownet foundations." Journal of Machine Learning Research 24.210 (2023): 1-55.
>
> > **(W2)** PLT introduces new hyperparameters ($\alpha$ and $M$).
>
> We agree with the reviewer that verifying the robustness of the reward mixing coefficient, $\alpha$, on more complex benchmarks. To explicitly demonstrate this, we conducted additional sensitivity analyses for $\alpha$ on biochemical discovery (TFBind8) and combinatorial optimization tasks (MIS). As shown in the table, PLT demonstrated consistently stable performance across a wide range of $\alpha$ values, with only minor variations in performance.
>
> **Ablation study for $\alpha$ on biochemical discovery and combinatorial optimization tasks.**
> || TFBind8 (Number of modes) | MIS (Set size) |
> |--|--|--|
> | $\alpha$=0.0            | 314 ± 2| 18.75 ± 0.16|
> | $\alpha$=1.0            | 320 ± 3| 18.67 ± 0.23|
> | $\alpha$=5.0            | 321 ± 1| 18.55 ± 0.15|
> | $\alpha$=10.0 (Default) | 319 ± 3| 19.03 ± 0.03|
> | $\alpha$=50.0           | 320 ± 1| 18.51 ± 0.23|
>
> > **(Q1)** How sensitive is the PLT mechanism in the paper to the choice of the backward policy $P_B$?
>
> As you mentioned, the choice of the backward policy $P_B$ is indeed a critical component in GFlowNet training dynamics.
>
> In our main experiments, we follow prior implementations:
> - Gridworld: a fixed uniform backward policy
> - Biochemical discovery and Combinatorial optimization: a learnable backward policy
>
> To verify the robustness of PLT to the choice of the backward policy, we additionally conducted experiments using a learnable backward policy in Gridworld.
>
> As shown in the table below, PLT consistently achieves the lowest L1 density error compared to other baseliens, demonstrating its robustness to the choice of backward policy $P_B$. We will include these new experimental results in our future manuscript.
>
> **Experiment results on Gridworld environment with learnable backward policy**
> |L1 ($\times10^{-5}$) | Sparse (d=2, H=128) | Asymmetric (d=2, H=128) |
> | -- | -- | -- |
> | Uniform   | 0.958 ± 0.239| 5.480 ± 3.648 |
> | PRT       | 1.174 ± 0.535| 4.284 ± 3.527 |
> | PER       | 1.330 ± 0.373| 2.866 ± 2.964 |
> | PLT (ours)| **0.858 ± 0.160**| **1.773 ± 0.547** |
>
> > **(Q2)** Could you provide theoretical insights or proofs to explain why the absolute loss difference specifically correlates with the optimal information gain or exploration metrics for GFlowNets?
>
> We appreciate your emphasis on theretical connection between learnability and optimal information gain. Unfortunately, providing a rigorous mathematical proof for this correlation presents significant challenges due to the highly non-linear dynamics of deep neural networks (Even PER does not have theoretical connection between information gain). We want to emphasize our method focuses on showing practical improvments across a diverse range of tasks, overcoming the severe limitations of prior baselines.
>
> > **(Q3)** The paper mentions in Section 5.3 that updating the reference policy too often or too infrequently (high or low update interval $M$) leads to suboptimal results or performance drops.
>
> For the update interval $M$, we observed that $M=100$ results in stable performance across all environments. Consequently, we fix $M$ as $100$ for all experiments we have done in this work. While there is a possibilty of dynamically adjust $M$, it may add complexity of the method and require manual tuning.

---

> > ### Author Rebuttal · Reviewer_MH45 · 2026-04-03
> >
> > The authors responded well to my questions. I will keep my original score, as my initial evaluation had already taken the paper’s corresponding contributions into account.

---

> > > ### Author Response · Authors · 2026-04-03
> > >
> > > We are glad to hear that our responses effectively addressed your questions. We will be sure to incorporate the additional analysis into the revised manuscript.

---

### Official Review · Reviewer_ZZEN · 2026-03-14

**Soundness:** 2
**Presentation:** 3
**Significance:** 3
**Originality:** 3
**Overall Recommendation:** 5
**Confidence:** 3

**Summary:**

The paper studies how to sample from the replay buffer during off-policy GFlowNet training. The authors argue that existing replay choices are problematic: uniform replay can over-focus on low-reward regions, reward-prioritized replay can induce mode collapse, and loss-prioritized replay can become unstable because GFlowNets only observe trajectory-level losses.

The key idea of the paper’s method is to define a terminal state’s “learnability” using the gap between the current policy’s trajectory loss and the loss under a lagged reference policy. They then sample replay states proportionally to a mixture of this learnability score and reward, with the goal of emphasizing newly discovered or forgotten regions while still favoring high-reward states.

Empirically, the paper provides evaluates on two synthetic gridworld variants, four biochemical discovery tasks (QM9, TFBind8, sEH, L14-RNA), and two combinatorial optimization tasks (MIS, MDS).

**Compliance With Llm Reviewing Policy:**

Affirmed.

**Final Justification:**

Rebuttal acknowledged questions.

**Key Questions For Authors:**

- Can you compare PLT against stronger off-policy GFlowNet baselines beyond Uniform/PRT/PER on the main biochemical tasks?
- Do you have any formal justification for the learnability score in Eq. (7), or any analysis showing that PLT improves optimization stability or variance relative to PER?

**Limitations:**

yes

**Strengths And Weaknesses:**

Strengths:

- The motivation is clearly tied to known failure modes in sparse and multimodal settings, and the method is specified concretely in its equations and algorithm.

  - The empirical evidence is reasonably broad in domain coverage: synthetic grids, biochemical discovery, and combinatorial optimization. The paper also includes sensitivity self-ablations in the appendix, plus robustness checks to another objective (DB) and another exploratory strategy (Teacher).

  - The paper's central idea appears novel at the replay-sampling level;p I at least know of no prior work introducing this exact learnability score based on the gap between current-policy and lagged-reference-policy losses for replay prioritization.

Weaknesses:

  - The paper provides almost no formal theory for the proposed prioritization rule. It defines learnability and introduces Eq. (8), but gives no theorem, proof, or derivation showing that PLT improves convergence, reduces variance, preserves the reward-matching objective, or yields an unbiased or better-conditioned gradient estimate. The method section moves directly from the definition to the algorithm and then to experiments.

  - The current evidence does not yet justify a strong claim of broad significance across off-policy GFlowNet methods. The paper mainly establishes that PLT is better than a narrow family of replay-sampling baselines, not that it is competitive with the strongest broader off-policy alternatives already in the literature. The paper's own related work mentions methods based on local search, intrinsic rewards, and auxiliary exploratory policies built from GFlowNets. Even if the focus of this work is purely on the replay buffer sampling, it’s important to understand performance compared to the current baselines.

---

> ### Author Rebuttal · Authors · 2026-03-31
>
> We sincerely thank the reviewer for the valuable feedback and acknowledging clear motivation and extensive experiments. We address your specific concerns as below:
>
> > **(W1 & Q2)** The paper provides almost no formal theory for the proposed prioritization rule.  / Do you have any formal justification for the learnability score in Eq. (7), or any analysis showing that PLT improves optimization stability or variance relative to PER?
>
> We acknowledge the reviewer's point that the paper does not present a formal proof of variance reduction or optimization stability. However, we respectfully note that deriving exact bounds on variance reduction and gradient estimates for deep neural networks with non-linear function approximators remains a highly non-trivial open challenge in the field.
>
> Rather than theoretical bounds, our work is motivated by the critical the empirical observation: prior sampling methods suffer from severe mode collapse or high variance, especially in large search spaces with sparse rewards. Our contribution is to provide a simple but effective solution for this problem, which still preserves GFlowNet's reward matching objective. As demonstrated across diverse benchmarks, from synthetic grids to biochemical discovery and combinatorial optimization, PLT consistently achieves better mode coverage compared to prior baselines.
>
> While we do not provide a novel convergence proof for PLT, we clarify that sampling distribution does not affect the unbiasedness of the TB/DB objectives thanks to its off-policyness nature [1,2].
>
> [1] Malkin, Nikolay, et al. "Trajectory balance: Improved credit assignment in gflownets." Advances in Neural Information Processing Systems 35 (2022): 5955-5967.
>
> [2] Bengio, Yoshua, et al. "Gflownet foundations." Journal of Machine Learning Research 24.210 (2023): 1-55.
>
> > **(W2 & Q1)** The current evidence does not yet justify a strong claim of broad significance across off-policy GFlowNet methods. / Can you compare PLT against stronger off-policy GFlowNet baselines beyond Uniform/PRT/PER on the main biochemical tasks?
>
> We would like to emphasize that PLT is an orthogonal improvement and therefore it functions seamlessly as a plug-and-play module that complements, rather than competes with advanced exploratory GFlowNet algorithms.
>
> To demonstrate this versatility, we have already verified PLT’s performance when combined with broad off-policy alternatives mentioned:
> - Local Search: In Figure 17,, we employ local search as the exploratory policy on biochemical tasks. We observed that PLT maintains superior mode coverage compared to baselines.
> - Auxiliary Exploratory Policies: In Figure 10, we augmented the Teacher policy with our method and other sampling methods. PLT consistently outperforms other sampling methods under this regime as well.

---

> > ### Author Rebuttal · Reviewer_ZZEN · 2026-04-03
> >
> > Thank you for pointing me to the additional figures, questions acknowledged and adjusting score accordingly.

---

> > > ### Author Response · Authors · 2026-04-03
> > >
> > > Thank you for your continued engagement and for adjusting your score. We will ensure that the clarifications to your questions are clearly integrated into the revised manuscript.

---

### Decision · Program_Chairs · 2026-04-30

**Decision:**

Reject

**Comment:**

While the problem addressed is interesting and important, there are several limitations:

* Lack of theoretical justification: The learnability metric is largely heuristic, with no formal guarantees on convergence, variance reduction (as also noted by reviewers ZZEN and MH45). Furthermore, the novelty of this term itself appeared to be limited.

* Limited baseline comparisons: Despite the broader ML literature on framing sampling as RL or optimization, the evaluation and comparison are mostly restricted to a narrow set of baselines in papers from Gflownet's authors. This makes it difficult to assess the method’s relative strength more broadly. It would be helpful to include, or at least discuss, related approaches from the wider literature outside the "Gflownet" community. For example, I recommend reading and citing papers like this (and many more papers citing these papers)

Zhang, Qinsheng, and Yongxin Chen. "Path integral sampler: a stochastic control approach for sampling."  ICLR 2022